# Sharp Representation Theorems for ReLU Networks with Precise Dependence on Depth

**Guy Bresler**
Department of EECS
MIT
Cambridge, MA 02139
guy@mit.edu

**Dheeraj Nagaraj**
Department of EECS
MIT
Cambridge, MA 02139
dheeraj@mit.edu

## Abstract

We prove sharp dimension-free representation results for neural networks with $D$ ReLU layers under square loss for a class of functions $\mathcal{G}_D$ defined in the paper. These results capture the precise benefits of depth in the following sense:

1. The rates for representing the class of functions $\mathcal{G}_D$ via $D$ ReLU layers is sharp up to constants, as shown by matching lower bounds.

2. For each $D$, $\mathcal{G}_D \subseteq \mathcal{G}_{D+1}$ and as $D$ grows the class of functions $\mathcal{G}_D$ contains progressively less smooth functions.

3. If $D' < D$, then the approximation rate for the class $\mathcal{G}_D$ achieved by depth $D'$ networks is strictly worse than that achieved by depth $D$ networks.

This constitutes a fine-grained characterization of the representation power of feedforward networks of arbitrary depth $D$ and number of neurons $N$, in contrast to existing representation results which either require $D$ growing quickly with $N$ or assume that the function being represented is highly smooth. In the latter case similar rates can be obtained with a single nonlinear layer. Our results confirm the prevailing hypothesis that deeper networks are better at representing less smooth functions, and indeed, the main technical novelty is to fully exploit the fact that deep networks can produce highly oscillatory functions with few activation functions.

## 1 Introduction

Deep neural networks are the workhorse of modern machine learning [1]. An important reason for this is the universal approximation property of deep networks which allows them to represent any continuous real valued function with arbitrary accuracy. Various representation theorems, establishing the universal approximation property of neural networks have been shown [2, 3, 4, 5, 6]. Under regularity conditions on the functions, a long line of work gives rates for approximation in terms of number of neurons [7, 8, 9, 10, 11, 12, 13, 14, 15, 16]. By now the case of a single layer of nonlinearities is fairly well understood, while the corresponding theory for deep networks is lacking.

Deep networks have been shown empirically to significantly outperform their shallow counterparts and a flurry of theoretical papers has aimed to understand this. For instance, [17] shows that letting depth scale with the number of samples gives minimax optimal error rates for non-parametric regression tasks. [18] considers hierarchical learning in deep networks, where training with SGD yields layers that successively construct more complex features to represent the function. While an understanding of the generalization performance of neural networks trained on data is a holy grail, motivated by the logic that expressivity of the network determines the fundamental barriers under arbitrary optimization procedures, in this paper we focus on the more basic question of representation.

A body of work on *depth separation* attempts to gain insight into the benefits of depth by constructing functions which can be efficiently represented by networks of a certain depth but cannot be represented by shallower networks unless their width is very large [19, 20, 12, 21, 22, 23, 24, 25]. For instance, [23] shows the existence of radial functions which can be easily approximated by two nonlinear layers but cannot be approximated by one nonlinear layer and [24] shows the existence of oscillatory functions which can be approximated easily by networks with $D^3$ nonlinear layers but cannot be approximated by $2^D$-width networks of $D$ nonlinear layers . In the different setting of representing probability distributions with sum-product networks, [26] shows strong $D$ versus $D + 1$ separation results. All of these results show *existence* of a function requiring a certain depth, but do not attempt to characterize the *class* of functions approximable by networks of a given depth.

For neural networks with $N$ nonlinear units in a single layer, classical results obtained via a law of large numbers type argument yields a $1/N$ rate of decay for the square loss [7, 8]. Several papers suggest a benefit of increased depth [9, 10, 12, 13] by implementing a Taylor series approximation to show that deep ReLU or RePU neural networks can achieve rates faster than $1/N$, when the function being represented is very smooth and the depth is allowed to grow as the loss tends to $0$. However, it was shown recently in [16] that when such additional smoothness is assumed, a *single* layer of nonlinearities suffices to achieve similar error decay rates. Therefore, the benefits of depth are not captured by these results.

The work most related to ours is [14], which considers representation of functions of a given modulus of continuity under the sup norm. When depth $D$ scales linearly with the total number of activation functions $N$, the rate of error decay is shown to be strictly better than when $D$ is held fixed. This does indicate that depth is fundamentally beneficial in representation, but the rates are dimension-dependent and hence, as will become clear, the results are far from sharply characterizing the exact benefits of depth.

In this paper we give a fine-grained characterization of the role of depth in representation power of ReLU networks. Given a network with $D$ ReLU layers and input dimension $d$, we define a class $\mathcal{G}_D$ of real valued functions characterized by the decay of their Fourier transforms, similar to the class considered in classical works such as [7]. As $D$ increases, the tails of the Fourier transforms are allowed to be fatter, thereby capturing a broader class of functions. Note that decay of a function's Fourier transform is well-known to be related to its smoothness (c.f., [27]). Our results stated in Section 4 show that a network with $N$ ReLU units in $D$ layers can achieve rates of the order $N^{-1}$ for functions in the class $\mathcal{G}_D$ whereas networks with $D' < D$ ReLU layers must suffer from slower rates of order $N^{-D'/D}$. All of these rates are optimal up to constant factors. As explained in Section 3, we prove these results by utilizing the compositional structure of deep networks to systematically produce highly oscillatory functions which are hard to produce using shallow networks.

**Comparison with some prior works.**   Based on comments from the reviewers, we discuss our results in comparison to [24] and [23]. [24] considers depth separation only and the results about oscillations produced networks of a certain depth found in that work is used to prove our lower bounds. We use such oscillations to prove representation theorems for general classes of functions and obtain explicit lower bounds on the squared error which match the error rates obtained for these classes of functions.

[23] concerns the representation of smooth radially symmetric functions. They use Fourier analysis and specifically Plancherel-Parseval formula to express square loss with respect to a specific distribution in terms how well the Fourier transform of the network approximates the Fourier transform of the target function. A radially symmetric function has a Fourier transform which is also radially symmetric, whereas any one dimensional function (such as the output of a ReLU function) has a Fourier transform supported in exactly one dimension when viewed as a tempered distribution. Therefore, they conclude using geometry of $\mathbb{R}^d$ that $e^{\Omega(d)}$ neurons are required to approximate such a function with vanishing error. We use Fourier transforms to obtain a 'one-dimensional' representation of the target function using the Fourier inversion formula.

**Organization.**   The paper is organized as follows. In Section 2, we introduce notation and define the problem. In Section 3, we overview the main idea behind our results, which are then stated formally in Section 4. Sections 5, 6, and 7 prove these results.

## 2 Notation, Problem Setup, and Fourier Norms

**Notation.** For $t \in \mathbb{R}$ let $\mathsf{ReLU}(t) = \max(0, t)$. In this work, the depth $D$ refers to the number of ReLU layers. Let the input dimension be $d$. Given $d_0, d_1, d_2, \ldots, d_D \in \mathbb{N}$, where $d_0 = d$. For $1 \leq i \leq D$, let $f_i : \mathbb{R}^{d_i} \to \mathbb{R}^{d_{i+1}}$ be defined by $f_i(x) = \sum_{j=1}^{d_i} \mathsf{ReLU}(\langle x, W_{ij} \rangle - T_{ij})e_j$, where $e_j$ are the standard basis vectors, $W_{ij} \in \mathbb{R}^{d_i}$, and $T_{ij} \in \mathbb{R}$. For $a \in \mathbb{R}^{d_D}$, we define the ReLU network corresponding to the parameters $d, D, d_1, \ldots, d_D, W_{ij}, T_{ij}$ and $a$ to be $\hat{f}(x) = \langle a, f_D \circ f_{D-1} \cdots \circ f_1(x) \rangle$. The number of ReLU units in this network is $N = \sum_{i=1}^{D} d_i$.

**The representation problem.** Consider a function $f : \mathbb{R}^d \to \mathbb{R}$. Given any probability measure $\mu$ over $B_d(r) := \{x \in \mathbb{R}^d : \|x\|_2 \leq r\}$, we want to understand how many ReLU units are necessary and sufficient in a neural network of depth $D$ such that its output $\hat{f}$ has square loss bounded as $\int \left( f(x) - \hat{f}(x) \right)^2 \mu(dx) \leq \epsilon$.

**Fourier norms.** Suppose $f$ has Fourier transform $F$, which for $f \in L^1(\mathbb{R}^d)$ is given by

$$F(\xi) = \int_{\mathbb{R}^d} f(x)e^{i\langle \xi, x \rangle} dx\,.$$

The Fourier transform is well-defined also for larger classes of functions than $L^1(\mathbb{R}^d)$ [27]. If $F$ is a function, then we assume $F \in L^1(\mathbb{R}^d)$, but we also allow it to be a finite complex measure and integration with respect to $F(\xi)d\xi$ is understood to be integration with respect to this measure. Under these conditions we have the Fourier inversion formula

$$f(x) = \frac{1}{(2\pi)^d} \int_{\mathbb{R}^d} F(\xi)e^{-i\langle \xi, x \rangle} d\xi\,. \tag{1}$$

For $\alpha \geq 0$, define the *Fourier norms* as in [7],

$$C_f^\alpha = \frac{1}{(2\pi)^d} \int_{\mathbb{R}^d} |F(\xi)| \cdot \|\xi\|^\alpha d\xi\,. \tag{2}$$

We define a sequence of function spaces $\mathcal{G}_K$ such that $\mathcal{G}_K \subseteq \mathcal{G}_{K+1}$ for $K \in \mathbb{N}$:

$$\mathcal{G}_K := \{f : \mathbb{R}^d \to \mathbb{R} : C_f^0 + C_f^{1/K} < \infty\}\,.$$

The domain $\mathbb{R}^d$ is usually implicit, but we occasionally write $\mathcal{G}_K(\mathbb{R}^d)$ or $\mathcal{G}_K(\mathbb{R})$ for clarity. Since decay of the Fourier transform is related to smoothness of the function, the sequence of function spaces $(\mathcal{G}_K)$ adds functions with less smoothness as $K$ increases. It is hard to exactly characterize this class of functions, but we note that a variety of function classes, when modified to decay to 0 outside of $B_d(r)$ (by multiplying with a suitable 'bump function'), are included in $\mathcal{G}_K$ for all $K$ large enough. These include polynomials, trigonometric polynomials, (for all $K \geq 1$) and any ReLU network of any depth (when $K \geq 2$). We also note that $\mathcal{G}_K$ contains uniformly continuous functions and dense in $L^p(\mathbb{R}^d)$ since they contain every Schwartz function. Based on reviewer feedback, we give a short sketch of the proof of uniform continuity in the appendix (Section B).

Theorems 1 and 2 show that the quantity $C_f^0(C_f^{1/K} + C_f^0)$ effectively controls the rate at which $f$ can be approximated by a depth $K$ ReLU network (at least in a ball of radius $r = 1$, with suitable modification for arbitrary $r$). As $K$ increases, the class of functions for which $C_f^0(C_f^{1/K} + C_f^0) \leq 1$ grows to include less and less smooth functions. The following two examples illustrate this behavior:

**Example 1.** *Consider the Gaussian function $f : \mathbb{R}^d \to \mathbb{R}$ given by $f(x) = e^{-\frac{\|x\|^2}{2}}$. Its Fourier transform is $F(\xi) = (2\pi)^{\frac{d}{2}} e^{-\frac{\|\xi\|^2}{2}}$. A simple calculation shows that $C_f^0(C_f^{1/K} + C_f^0) \sim d^{1/2K} + 1$. Thus, when $K \gtrsim \log d$, the quantity $C_f^0(C_f^{1/K} + C_f^0)$ remains bounded for any dimension $d$.*

**Example 2.** *Let $n \in \mathbb{N}$ be large and consider the function $f : \mathbb{R} \to \mathbb{R}$ given by $f(x) = \cos(nx)/n^\alpha$. As $\alpha$ decreases, the oscillations grow in magnitude and one can check that $C_f^0(C_f^{1/K} + C_f^0) = n^{1/K-2\alpha} + n^{-2\alpha}$. When $K \geq 1/2\alpha$, the rates are essentially independent of $n$.*

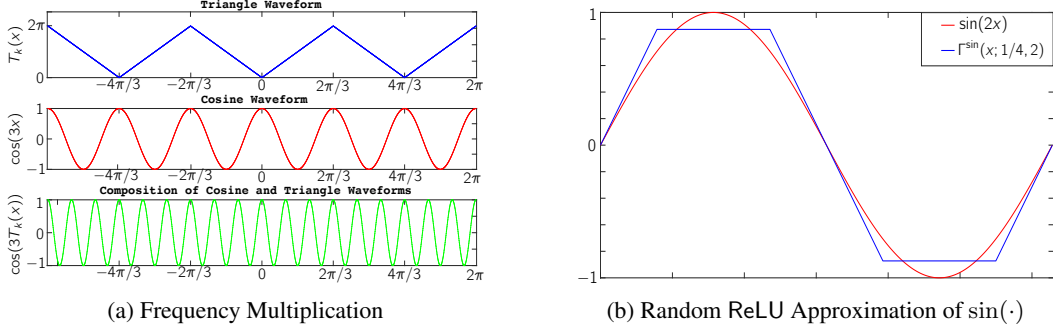

(a) Frequency Multiplication          (b) Random ReLU Approximation of $\sin(\cdot)$

Figure 1: Conceptually, our representation result is shown by a combination of composition of ReLU with sinusoids to boost frequencies as depicted in (a), and using ReLUs to approximate sinusoids which appear in the Fourier transform of a function as depicted in (b).

## 3 Using Depth to Improve Representation

Before stating our representation theorems in the next section, we now briefly explain the core ideas:

1. Following [7], we use the inverse Fourier transform to represent $f(x)$ as an expectation of $A\cos(\langle\xi,x\rangle+\theta(\xi))$ for some random variable $\xi$ and then implement $\cos(\langle\xi,x\rangle+\theta(\xi))$ using ReLU units.

2. We use an idea similar to the one in [24] to implement a triangle waveform $T_k$ with $2k$ peaks using $\sim k^{1/D}$ ReLU units arranged in a network of depth $D$.

3. Composition of low frequency cosines with triangle waveforms is then used to efficiently approximate the high frequency cosines of the form $\cos(\langle\xi,x\rangle+\theta(\xi))$ via ReLU units.

Suppose that we want to approximate the function $f(t)=\cos(\omega t)$ for $t$ in the interval $[-1,1]$ using a ReLU network with a single hidden layer (as in Item 1 above). Because the interval $[-1,1]$ contains $\Omega(\omega)$ periods of the function, effectively tracking it requires $\Omega(\omega)$ ReLU units. It turns out that this dependence can be significantly improved if we allow two nonlinear layers. The first layer is used to implement the triangle waveform $T_k$ on $[-1,1]$ for $k=\Theta(\sqrt{\omega})$, which oscillates at a frequency $\sqrt{\omega}$ and uses $O(\sqrt{\omega})$ ReLU units. Then the second layer is used to implement $\cos(\sqrt{\omega}t)$, again with $O(\sqrt{\omega})$ ReLU units. The output of the two layers combined is $\cos(\sqrt{\omega}T_k(t))$, and since $\cos(\sqrt{\omega}t)$ and $T_k(t)$ each oscillate with frequency $\sqrt{\omega}$, it follows that their composition $\cos(\sqrt{\omega}T_k(t))$ oscillates at the frequency $\omega$, and one can show more specifically that we obtain the output $\cos(\omega t)$. We check that the network requires only $O(\sqrt{\omega})$ ReLU units. A similar argument shows that networks of depth $D$ require only $O(\omega^{1/D})$ ReLU units. Surprisingly, this simple idea yields optimal dependence of representation power on depth. We illustrate this in Figure 1a.

## 4 Main Results

**Theorem 1.** *Suppose $f:\mathbb{R}^d\to\mathbb{R}$ is such that $f\in\mathcal{G}_K$ for some $K\geq 1$. Let $\mu$ be any probability measure over $B_d(r)$ and let $D\in\mathbb{N}$ be such that $D\leq K$. There exists a* ReLU *network with $D$* ReLU *layers and at most $N_0$* ReLU *units in total such that its output $\hat{f}$ satisfies*

$$\int \big(f(x)-\hat{f}(x)\big)^2\mu(dx)\leq A_0\Big(\frac{D}{N_0}\Big)^{D/K}\big(r^{1/K}C_f^{1/K}C_f^0+(C_f^0)^2\big), \qquad (3)$$

*where $A_0$ is a universal constant given in the proof.*

We give the proof in Section 6.

**Remark 1.** *Theorem 1 requires $D\leq K$, but can be applied also in the case $D>K$ as follows. If $D>K$ and $f\in\mathcal{G}_K$, then $f\in\mathcal{G}_D$ since $C_f^{1/D}+C_f^0\leq 2(C_f^{1/K}+C_f^0)$. By an application of Theorem 1 for $f\in\mathcal{G}_D$, we obtain an upper bound on the square loss of $O\big(\frac{D}{N_0}\big(r^{1/D}C_f^{1/D}C_f^0+(C_f^0)^2\big)\big)$, which is of the order $1/N_0$. Thus, our upper bound for the class $\mathcal{G}_K$ becomes better as the depth $D$ increases and saturates at $D=K$, giving an upper bound of the order $1/N_0$.*

**Remark 2.** *Getting rates faster than $1/N_0$ for $f \in \mathcal{G}_K$ using $D > K$ layers may be possible using ideas from [16]. We intend to address this problem in future work.*

**Remark 3.** *When $D = K = 1$, Theorem 1 captures the $O(1/N_0)$ rate achieved by [8] which is shown under the assumption that $C_f^2 < \infty$ instead of $C_f^1 + C_f^0 < \infty$ as given here.*

We next give a matching lower bound in Theorem 2, which also shows depth separation between depth $D$ and $D + 1$ networks for arbitrary $D$.

**Theorem 2.** *Let $K \geq 1$ be fixed, $D \leq K$, and $r > 0$. Let $\mu$ be the uniform measure over $[-r, r]$. There exists $f : \mathbb{R} \to \mathbb{R}$ in $\mathcal{G}_K(\mathbb{R})$ and a universal constant $B_0$ such that for any $\hat{f} : \mathbb{R} \to \mathbb{R}$ given by the output of a $D$ layer ReLU network with at most $N_0$ nonlinear units,*

$$\int \big(f(x) - \hat{f}(x)\big)^2 \mu(dx) \geq B_0\big(r^{1/K} C_f^{1/K} C_f^0 + (C_f^0)^2\big)\Big(\frac{D}{N_0}\Big)^{D/K}.$$

The proof, given in Section 7, is based on the fact that the output of a $D$ layer ReLU network with $N_0$ units can oscillate at most $\sim (N_0/D)^D$ times ([24]). Therefore, such a network cannot capture the oscillations in $\cos(\omega x)$ whenever $\omega > (N_0/D)^D$.

**Remark 4.** *Theorems 1 and 2 together recover the depth separation results of [24] for the case of ReLU networks. Following reviewer feedback, we would like to note that these lower bounds are in spirit different from the ones found in [24]. We have restricted the lower bounds to the one-dimensional case - which contains the functions which are easiest to represent. We can consider higher dimensional versions by considering functions of the form $f(\langle a, x \rangle)$ for some $f : \mathbb{R} \to \mathbb{R}$ for arbitrary $a \in \mathbb{R}^d$ and the lower bounds follow.*

## 5 Technical Results

Before delving into the proof of Theorems 1 and 2, we require some technical lemmas.

### 5.1 Triangle Waveforms

Consider the triangle function parametrized by $\alpha, \beta > 0$ $T(\,\cdot\,; \alpha, \beta) : \mathbb{R} \to \mathbb{R}$ defined by

$$T(t; \alpha, \beta) = \begin{cases} \beta t & \text{if } t \in [0, \alpha] \\ 2\alpha\beta - \beta t & \text{if } t \in (\alpha, 2\alpha] \\ 0 & \text{otherwise.} \end{cases} \tag{4}$$

We construct the triangle waveform, parametrized by $\alpha, \beta > 0$ and $k \in \mathbb{N}$, defined as

$$T_k(t; \alpha, \beta) = \sum_{i=-k+1}^{k} T(t - 2\alpha(i-1); \alpha, \beta). \tag{5}$$

The triangle waveform $T_k(\,\cdot\,; \alpha, \beta)$ is supported over $[-2\alpha k, 2\alpha k]$ and can be implemented with $4k + 1$ ReLUs in a single layer. We state the following basic results and refer to Appendix A.1 for their proofs.

**Lemma 1** (Symmetry). *The triangle waveform $T_k$ satisfies the following symmetry properties:*

   *1. Let $t \in [2m\alpha, 2(m+1)\alpha]$ for some $-k \leq m \leq k - 1$. Then*

$$T_k(t; \alpha, \beta) = T_k(t - 2m\alpha; \alpha, \beta) = T(t - 2m\alpha; \alpha, \beta).$$

   *2. Let $t \in [0, 2k\alpha]$. Then, $T_k(2k\alpha - t; \alpha, \beta) = T_k(t; \alpha, \beta)$.*

**Lemma 2** (Composition). *Let $l \in \mathbb{N}$. If $\alpha\beta = 2al$, then for every $t \in \mathbb{R}$, $T_l(T_k(t; \alpha, \beta); a, b) = T_{2kl}(t; \frac{a}{\beta}, b\beta)$.*

Lemma 2 shows that when we compose two triangle wave forms of the right height and width, their frequencies multiply. For instance, this implies that $T_{2kl}$ can be implemented with $O(k + l)$ ReLUs in two layers instead of $O(kl)$ ReLUs in a single layer.

## 5.2 Representing Sinusoids with Random ReLUs

Define $R_4(t; S) := \mathsf{ReLU}(t) + \mathsf{ReLU}(t - \frac{\pi}{\omega}) - \mathsf{ReLU}\left(t - \frac{\pi S}{\omega}\right) - \mathsf{ReLU}\left(t - \frac{\pi(1-S)}{\omega}\right)$ for $t \in \mathbb{R}$ and $S \in [0, 1]$. Henceforth let $S \sim \mathsf{Unif}[0, 1]$. We let

$$\Gamma^{\sin}(t; S, \omega) := \frac{\pi \omega}{2} \sin(\pi S)[R_4(t; S, \omega) - R_4(t - \tfrac{\pi}{\omega}; S, \omega)].$$

and

$$\Gamma_n^{\cos}(t; S, \omega) := \sum_{i=-n-1}^{n} \Gamma^{\sin}(t - \tfrac{2\pi i}{\omega} + \tfrac{\pi}{2\omega}; S, \omega).$$

for some $n \in \mathbb{N}$ to be set later. We refer to Figure 1b for an illustration of the function $\Gamma^{\sin}(\,\cdot\,; S, \omega)$.

**Lemma 3.** $\Gamma_n^{\cos}(t; S, \omega)$ *satisfies the following properties:*

1. *For every* $-\frac{\pi}{\omega} - n\frac{2\pi}{\omega} \leq t \leq n\frac{2\pi}{\omega} + \frac{\pi}{\omega}$, $\mathbb{E}\Gamma_n^{\cos}(t; S, \omega) = \cos(\omega t)$.

2. *For every* $t \in \mathbb{R}$, $|\Gamma_n^{\cos}(t; S, \omega)| \leq \pi^2/4$ *almost surely.*

3. $\Gamma_n^{\cos}(\,\cdot\,; S, \omega)$ *can be implemented using* $16n + 16$ $\mathsf{ReLU}$ *units in a single layer. (With a bit more care this can be improved to* $8n + 10$, *but we ignore this for the sake of simplicity.)*

The proof, deferred to Appendix A.2, is based on a simple application of integration by parts.

## 5.3 Frequency Multipliers

The lemma below combines the considerations from Section 5.1 to show that composing a ReLU estimator for a low frequency cosine function with a low frequency triangle waveform gives an estimator for a high frequency cosine function as described in Section 3.

**Lemma 4.** *Recall the triangle waveform* $T_k(\,\cdot\,; \alpha, \beta)$ *defined above. Fix* $\beta > 1$ *and* $\omega > 1$. *Set* $\alpha = (2n+1)\pi/\beta\omega$ *and* $k = \lceil (r + \frac{\pi}{\beta\omega})/2\alpha \rceil$. *For any* $\theta \in [-\pi, \pi]$ *and* $t \in [-r, r]$ *we have that*

$$\mathbb{E}\Gamma_n^{\cos}(T_k(t + \tfrac{\theta}{\beta\omega}; \alpha, \beta); S, \omega) = \cos(\beta\omega t + \theta).$$

The proof of this lemma follows from Lemma 3, using the fact that the triangle waveform $T_k$ makes the waveform repeat multiple times. The formal proof is given in Appendix A.2.

# 6 Proof of Theorem 1

We want to approximate $f \in \mathcal{G}_K$ using a depth $D$ neural network with $D \leq K$. Let $F$ be the Fourier transform of $f$, where $F(\xi) = |F(\xi)|e^{-i\theta(\xi)}$ for some $\theta(\xi) \in [-\pi, \pi]$ (such a choice exists via Jordan decomposition and the Radon-Nikodym Theorem). Then, by the Fourier inversion formula,

$$f(x) = \frac{1}{(2\pi)^d} \int_{\mathbb{R}^d} F(\xi) e^{-i\langle \xi, x \rangle} d\xi = \frac{1}{(2\pi)^d} \int_{\mathbb{R}^d} |F(\xi)| e^{-i[\langle \xi, x \rangle + \theta(\xi)]} d\xi$$

$$= \frac{1}{(2\pi)^d} \int_{\mathbb{R}^d} |F(\xi)| \cos\left(\langle \xi, x \rangle + \theta(\xi)\right) d\xi = C_f^0 \mathbb{E}_{\xi \sim \nu_f} \cos\left(\langle \xi, x \rangle + \theta(\xi)\right). \quad (6)$$

Here $\nu_f$ is the probability measure given by $\nu_f(d\xi) = \frac{|F(\xi)|d\xi}{(2\pi)^d C_f^0}$. We start with the case $D = 1$.

## 6.1 D = 1

**Cosine as an expectation over** $\mathsf{ReLU}$s. Suppose $\xi \neq 0$. Since $x \in B_d(r)$, we know that $\frac{\langle x, \xi \rangle}{\|\xi\|} \in [-r, r]$. Let $n = \lceil \|\xi\| r/2\pi \rceil$, $\omega = \|\xi\|$, and $t = \frac{\langle \xi, x \rangle}{\|\xi\|} + \frac{\theta(\xi)}{\|\xi\|}$ in Item 1 of Lemma 3 to conclude that

$$\cos(\langle \xi, x \rangle + \theta(\xi)) = \cos(\|\xi\| \tfrac{\langle \xi, x \rangle}{\|\xi\|} + \theta(\xi)) = \mathbb{E}_{S \sim \mathsf{Unif}[0,1]} \Gamma_n^{\cos}\left(\tfrac{\langle \xi, x \rangle}{\|\xi\|} + \tfrac{\theta(\xi)}{\|\xi\|}; S, \|\xi\|\right). \quad (7)$$

By Item 3 of Lemma 3, the unbiased estimator $\Gamma_n^{\cos}\left(\frac{\langle\xi,x\rangle}{\|\xi\|}+\frac{\theta(\xi)}{\|\xi\|};S,\|\xi\|\right)$ for $\cos(\langle\xi,x\rangle+\theta(\xi))$ can be implemented using $16n+16$ ReLUs.

If $\xi=0$, then $\cos(\langle\xi,x\rangle+\theta(\xi))$ is a constant and 2 ReLUs suffice to implement this function over $[-r,r]$. Thus, for any value of $\xi$ we can implement an unbiased estimator for $\cos(\langle\xi,x\rangle+\theta(\xi))$ using $16n+16$ ReLUs (note that this quantity depends on $\|\xi\|$ through $n$). We call this unbiased estimator $\hat{\Gamma}(x;\xi,S)$. By Item 2 of Lemma 3, $|\hat{\Gamma}(x;\xi,S)|\leq\pi^2/4$ almost surely.

**Obtaining an estimator via random sampling.** Let $(\xi,S)\sim\nu_f\times\mathsf{Unif}([0,1])$. For every $x\in B_d(r)$, (6) and (7) together imply that $f(x)=C_f^0\mathbb{E}\hat{\Gamma}(x;\xi,S)$. We construct an empirical estimate by sampling $(\xi_j,S_j)\sim\nu_f\times\mathsf{Unif}([0,1])$ i.i.d. for $j=1,2,\ldots,l$, yielding

$$\hat{f}(x)=\frac{1}{l}\sum_{j=1}^{l}C_f^0\hat{\Gamma}(x;\xi_j,S_j).\tag{8}$$

By Fubini's Theorem and the fact that $\hat{\Gamma}(x;\xi_j,S_j)$ are i.i.d., for any probability measure $\mu$ we have

$$\mathbb{E}\int\left(f(x)-\hat{f}(x)\right)^2\mu(dx)=\int\mathbb{E}\left(f(x)-\hat{f}(x)\right)^2\mu(dx)\leq(C_f^0)^2\sup_x\frac{\mathbb{E}|\hat{\Gamma}(x;\xi_j,S_j)|^2}{l}\leq\frac{(C_f^0)^2\pi^4}{16l}.$$

In the last step we have used the fact that $|\hat{\Gamma}(x;\xi,S)|\leq\pi^2/4$ almost surely. By Markov's inequality, with probability at least $2/3$ (over the randomness in $(\xi_j,S_j)_{j=1}^l$) we have

$$\int\left(f(x)-\hat{f}(x)\right)^2\mu(dx)\leq 3(C_f^0)^2\pi^4/16l.\tag{9}$$

Let the number of ReLU units used to implement $\hat{\Gamma}(\cdot;\xi_j,S_j)$ be $N_j$. Note that $N_j$ is a random variable bounded depending on the random $\xi_j$ according to $N_j\leq 8\|\xi_j\|r/\pi+32$. Therefore the total number of ReLUs used is $N=\sum_{j=1}^l N_j\leq\sum_{i=1}^l(8\|\xi_i\|r/\pi+32)$. By Minkowski's inequality, for every $K\geq 1$, $N^{1/K}\leq\sum_{i=1}^l(8\|\xi_i\|r/\pi)^{1/K}+(32)^{1/K}$, so by definition of $\nu_f$ and the Fourier norms, $\mathbb{E}N^{1/K}\leq l\left[\left(\frac{8r}{\pi}\right)^{1/K}\frac{C_f^{1/K}}{C_f^0}+(32)^{1/K}\right]$. Markov's inequality now implies that with probability at least $1/2$, the number of ReLUs used is bounded as

$$N^{1/K}\leq 2l\left[\left(\frac{8r}{\pi}\right)^{1/K}\frac{C_f^{1/K}}{C_f^0}+(32)^{1/K}\right]:=l\cdot D_0.\tag{10}$$

**Combining the two bounds.** Let $D_0$ be as defined in (10) just above. Given $N_0\in\mathbb{N}$ such that $N_0\geq D_0^K$, we take $l=\lfloor N_0^{1/K}/D_0\rfloor$. By the union bound, both Equations (9) and (10) must hold with positive probability, so there exists a configuration with $N$ ReLUs such that $N\leq N_0$ and

$$\int\left(f(x)-\hat{f}(x)\right)^2\mu(dx)\leq\frac{1}{N_0^{1/K}}\left(6\pi^4 C_f^0 C_f^{1/K}r^{1/K}+8\pi^4(C_f^0)^2\right).$$

If $N_0\leq D_0^K$ we just use a network that always outputs 0. From Equation (6), we see that $|f(x)|\leq C_f^0$ for every $x$, so the last displayed equation holds (up to a constant factor) also in this case.

## 6.2  D > 1

We follow the same overall procedure as the $D=1$ case, but we will use the frequency multiplication technique to implement the cosine function with fewer ReLU units. For each $\xi$, we want an unbiased estimator for $\cos(\|\xi\|t+\theta(\xi))$ for $t\in[-r,r]$ and $\theta(\xi)\in[-\pi,\pi]$. Assume $\xi\neq 0$.

**Triangle waveform.** In Lemma 4 we take $\omega=\|\xi\|^{1/D}$, $n=\lceil(\|\xi\|r)^{1/D}/2\pi\rceil$, $\beta=\|\xi\|^{1-1/D}$, and let $\alpha$ and $k$ be determined as given in the statement of the Lemma i.e. $\alpha=(2n+1)\pi/\|\xi\|$ and $k\geq\lceil(r+\frac{\pi}{\|\xi\|})/2\alpha\rceil$. Note that $\Gamma_n^{\cos}$ can be implemented by the $D$th ReLU layer and therefore it is sufficient to implement $T_k(\cdot;\alpha,\|\xi\|^{1-1/D})$ using the previous $D-1$ ReLU layers as follows.

Let $l := \left\lceil \frac{1}{2}(r\|\xi\|)^{1/D} \right\rceil$ and $\gamma := \|\xi\|^{1/D}$. Let $\alpha_1 := (2l)^{D-2}(2n+1)\pi/\|\xi\|$ and for $i = 2 \ldots, D-1$, $\alpha_i := \alpha_1 \left(\gamma/2l\right)^{i-1}$. For $i = 1, \ldots, D-1$, we define $f_i : \mathbb{R} \to \mathbb{R}$ as $f_i(t) = T_l(t; \alpha_i, \gamma)$. Clearly, $f_{D-1} \circ \cdots \circ f_1(t)$ can be implemented by a depth $D-1$ ReLU network by implementing the function $f_i$ using the $i$th ReLU layer. It now follows by a straightforward induction using Lemma 2 that

$$f_{D-1} \circ \cdots \circ f_1(t) = T_k(t; \alpha, \|\xi\|^{1-1/D}),$$

where $k = 2^{D-2} l^{D-1} \geq \lceil (r + \frac{\pi}{\|\xi\|})/2\alpha \rceil$.

Each of the first $D-1$ layers require $4l + 1$ ReLU units and by Item 3 of Lemma 3, the $D$th layer requires $16n + 16$ ReLU units. If the number of ReLUs used is $N(\xi)$, then

$$N(\xi) \leq (D-1)(4l+1) + 16n + 16 \leq \left(\tfrac{8}{\pi} + 2D - 2\right)(r\|\xi\|)^{1/D} + 5D + 27.$$

**Estimator via sampling.** As in the $D = 1$ case, we form the estimator in (8) by sampling $(\xi_j, S_j)$ i.i.d. from the distribution $\nu_f \times \mathsf{Unif}([0,1])$, but will now implement the cosines using a $D$ layer ReLU network as described above. This uses $N \leq \sum_{i=1}^{l} N(\xi_i)$ nonlinear units. Let $K \geq D$. Applying Minkowski's inequality to $N^{D/K}$ as in the $D = 1$ case, we obtain that

$$\mathbb{E} N^{D/K} \leq l \left[ \left(\tfrac{8}{\pi} + 2D - 2\right)^{D/K} r^{1/K} \frac{C_f^{1/K}}{C_f^0} + (5D + 27)^{D/K} \right].$$

Equation (9) remains the same in this case too. Therefore, using the union bound just like in the case $D = 1$, we conclude that there exists a configuration of weights for a $D$ layer ReLU network with $N \leq N_0$ ReLU units for any $N_0 \in \mathbb{N}$ such that

$$\int \left(f(x) - \hat{f}(x)\right)^2 \mu(dx) \leq \frac{3\pi^4 (2D+1)^{D/K}}{4N_0^{D/K}} r^{1/K} C_f^{1/K} C_f^0 + \frac{3\pi^4 (5D+27)^{D/K}}{4N_0^{D/K}} (C_f^0)^2.$$

## 7  Proof of Theorem 2

We will exhibit a function $f$ which is challenging to estimate over $[-r, r]$ by $\hat{f}$ with respect to square loss over the uniform measure $\mu$.

We use the idea of *crossing numbers* from [24] for the lower bounds. Given a continuous function $\hat{f} : \mathbb{R} \to \mathbb{R}$, let $\mathcal{I}_{\hat{f}}$ denote the partition of $\mathbb{R}$ into intervals where $\mathbb{1}(\hat{f} \geq 1/2)$ is a constant, and define $\mathsf{Cr}(\hat{f}) = |\mathcal{I}_{\hat{f}}|$. Let the set of endpoints of intervals in $\mathcal{I}_{\hat{f}}$ be denoted by $\mathcal{N}_{\hat{f}}$. Clearly, $|\mathcal{N}_{\hat{f}}| \leq \mathsf{Cr}(\hat{f})$.

Consider the function $f : \mathbb{R} \to \mathbb{R}$ given by $f(x) = \frac{1 + \cos(\frac{\omega x}{r})}{2\omega^\alpha}$ for some $\alpha > 0$ and $\omega > 1$. The Fourier transform of $f$ is the measure $\frac{\pi}{2\omega^\alpha} \left[ 2\delta(\xi) + \delta(\xi - \frac{\omega}{r}) + \delta(\xi + \frac{\omega}{r}) \right]$. Integrating yields $C_f^0 = 1/\omega^\alpha$ and $C_f^{1/K} = \frac{1}{2} r^{-1/K} \omega^{1/K - \alpha}$. We will later choose $\alpha = 1/2K$, for which it follows that $1/2 \leq C_f^0 C_f^{1/K} r^{1/K} + (C_f^0)^2 \leq 3/2$. We first show the following basic lemma.

**Lemma 5.** *Let $\omega = 2\pi L$ for some $L \in \mathbb{N}$. If $\mathsf{Cr}(\omega^\alpha \hat{f}) < 2L$ then*

$$\frac{1}{2r} \int_{-1}^{1} \left(f(x) - \hat{f}(x)\right)^2 dx \geq \frac{\pi}{16} \frac{2L - \mathsf{Cr}(\omega^\alpha \hat{f})}{\omega^{2\alpha+1}}.$$

*Proof.* Partition the interval $[-r, r)$ into $2L$ subintervals of the form $[ir/L, r(i+1)/L)$ for $-L \leq i \leq L - 1$. By a simple counting argument, there exist at least $2L - |\mathcal{N}_{\omega^\alpha \hat{f}}| \geq 2L - \mathsf{Cr}(\omega^\alpha \hat{f})$ such intervals which do not contain any point from the set $\mathcal{N}_{\omega^\alpha \hat{f}}$. Let $[ir/L, r(i+1)/L)$ be such an interval. Then either 1) $\hat{f}(x) \geq 1/2\omega^\alpha$ for every $x \in [ir/L, r(i+1)/L)$, or 2) $\hat{f}(x) < 1/2\omega^\alpha$ for

every $x \in [ir/L, r(i+1)/L)$. Without loss of generality, suppose that the first of these is true. Then

$$
\frac{1}{2r} \int_{ri/L}^{r(i+1)/L} (f(x) - \hat{f}(x))^2 dx = \frac{1}{2r} \int_{ri/L}^{r(i+1)/L} \left( \frac{1 + \cos(2\pi Lx/r)}{2\omega^\alpha} - \hat{f}(x) \right)^2 dx
$$

$$
\geq \frac{1}{2r} \int_{(4i+1)r/4L}^{(4i+3)r/4L} \left( \frac{\cos(2\pi Lx/r)}{2\omega^\alpha} + \frac{1}{2\omega^\alpha} - \hat{f}(x) \right)^2 dx \geq \frac{1}{2r} \int_{(4i+1)r/4L}^{(4i+3)r/4L} \left( \frac{\cos(2\pi Lx/r)}{2\omega^\alpha} \right)^2 dx
$$

$$
= \frac{1}{8r\omega^{2\alpha}} \int_{(4i+1)r/4L}^{(4i+3)r/4L} \cos^2(2\pi Lx/r) dx = \frac{\pi}{16\omega^{2\alpha+1}} \ . \tag{11}
$$

In the second step we have used the fact that $\cos(\frac{2\pi L}{r}x) \leq 0$ for $\frac{ir}{L} + \frac{r}{4L} \leq x \leq \frac{ir}{L} + \frac{3r}{4L}$. Adding the contributions to the integral in the statement of the lemma over the collection of such intervals which do not contain any point from $\mathcal{N}_{\hat{f}}$ yields the result. $\qquad \square$

We now refer to Lemma 3.2 in [24], which states that because $\hat{f}$ is the output of a $D$ layer ReLU network with at most $N_0$ ReLUs, its crossing number is bounded as $|\mathsf{Cr}(\hat{f})| \leq 2 (2N_0/D)^D$. Taking $L = \lceil 2 (2N_0/D)^D \rceil$ in Lemma 5 and recalling that $\omega = 2\pi L$, we obtain

$$
\frac{1}{2r} \int_{-r}^{r} \left( f(x) - \hat{f}(x) \right)^2 dx \geq \frac{\pi}{32\omega^{2\alpha}} = \frac{\pi}{32(2\pi)^{2\alpha}L^{2\alpha}} \geq \frac{\pi D^{2\alpha D}}{32(6\pi)^{2\alpha}(2N_0)^{2\alpha D}} \ .
$$

Recall from just before Lemma 5 that for $\alpha = 1/2K$ we have $1/2 \leq C_f^0 C_f^{1/K} r^{1/K} + (C_f^0)^2 \leq 3/2$. Thus, from the last displayed equation we conclude that there is a universal constant $B_0$ such that for any $\hat{f}$ that is the output of a $D$ layer $N_0$ unit ReLU network, the squared loss in representing $f$ is

$$
\frac{1}{2r} \int_{-r}^{r} \left( f(x) - \hat{f}(x) \right)^2 dx \geq B_0 \left( C_f^0 C_f^{1/K} r^{1/K} + (C_f^0)^2 \right) \left( \frac{D}{N_0} \right)^{D/K} \ .
$$

This completes the proof of Theorem 2.

## Broader Impact

This section is not applicable to this work.

## Acknowledgments and Disclosure of Funding

This work was supported in part by MIT-IBM Watson AI Lab and NSF CAREER award CCF-1940205

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
