[Supplementary Material]

# A  Supplementary Material

## A.1  Frequency Multipliers - Proofs

We skip the proof of Lemma 1 since it is elementary. We give a proof of Lemma 2 as follows.

*Proof of Lemma 2.* $T_k(t; \alpha, \beta)$ has $4k$ straight line segments which either increase from 0 to $\alpha\beta = 2al$ or decrease from $\alpha\beta$ to 0. For each of these line segments, the entire set of values of $T_l(\,\cdot\,; a, b)$ in $[0, 2al]$ is repeated once. This gives us $4kl$ triangles. The height of these triangles is the same as that of $T_l(\,\cdot\,; a, b)$ which is $ab$. The domain of the triangle waveform is the same as that of $T_k(\,\cdot\,; \alpha, \beta)$, which is $[-2k\alpha, 2k\alpha]$. From this we conclude the statement of the lemma. $\qquad\square$

## A.2  ReLU Representation for Sinusoids - Proofs

Let $\omega > 0$. We want to represent $t \to \sin(\omega t)$ for $t \in [0, \pi/\omega]$ in terms of ReLU functions. The first part of the argument entails manipulation of an integral, and then the resulting identity will be applied to obtain the proofs of Lemmas 3 and 4.

To start, integration by parts yields

$$\int_0^{\pi/\omega} \omega^2 \sin(\omega T)\mathsf{ReLU}(t - T)dT = \omega t - \sin(\omega t)\,.$$

Replacing $t$ with $\pi/\omega - t$, we have

$$\int_0^{\pi/\omega} \omega^2 \sin(\omega T)\mathsf{ReLU}(\pi/\omega - t - T)dT = \pi - \omega t - \sin(\omega t)\,,$$

and adding the last two equations gives

$$\int_0^{\pi/\omega} \omega^2 \sin(\omega T) \left[\mathsf{ReLU}(\pi/\omega - t - T) + \mathsf{ReLU}(t - T)\right] dT = \pi - 2\sin(\omega t)\,.$$

From the case $t = 0$ in the last equation, we conclude that

$$\pi = \int_0^{\pi/\omega} \omega^2 \sin(\omega T)\left[\pi/\omega - T\right] dT\,.$$

Combining the last two equations, we obtain the identity

$$\sin(\omega t) = \frac{1}{2}\int_0^{\pi/\omega} \omega^2 \sin(\omega T)\left[\pi/\omega - T - \mathsf{ReLU}(\pi/\omega - t - T) - \mathsf{ReLU}(t - T)\right] dT\,.$$

Making the transformation $S = \frac{T\omega}{\pi}$, the integral can be rewritten as

$$\sin(\omega t) = \frac{\pi}{2}\int_0^1 \omega \sin(\pi S)\left[\frac{\pi}{\omega}(1 - S) - \mathsf{ReLU}\left(\frac{\pi}{\omega}(1 - S) - t\right) - \mathsf{ReLU}\left(t - \frac{\pi S}{\omega}\right)\right] dS\,. \quad (12)$$

Now recall the function $R_4(\,\cdot\,; S, \omega)$ as defined in Section 5.2. A simple calculation shows that

$$R_4(t; S, \omega) = \begin{cases} 0 & \text{if } t \notin [0, \frac{\pi}{\omega}] \\ \frac{\pi}{\omega}(1 - S) - \mathsf{ReLU}\left(\frac{\pi}{\omega}(1 - S) - t\right) - \mathsf{ReLU}\left(t - \frac{\pi S}{\omega}\right) & \text{if } t \in [0, \frac{\pi}{\omega}]\,, \end{cases}$$

so if we let $S$ be a random variable with $S \sim \mathrm{Unif}([0, 1])$ we can rewrite (12) as

$$\mathbb{E}\frac{\pi\omega}{2}\sin(\pi S)R_4(t; S, \omega) = \begin{cases} 0 & \text{if } t \notin [0, \frac{\pi}{\omega}] \\ \sin(\omega t) & \text{if } t \in [0, \frac{\pi}{\omega}]\,. \end{cases}$$

It then follows that

$$\mathbb{E}\frac{\pi\omega}{2}\sin(\pi S)[R_4(t; S, \omega) - R_4(t - \tfrac{\pi}{\omega}; S, \omega)] = \begin{cases} 0 & \text{if } t \notin [0, \frac{\pi}{\omega}] \\ \sin(\omega t) & \text{if } t \in [0, \frac{2\pi}{\omega}]\,. \end{cases} \quad (13)$$

*Proof of Lemma 3.* The first item follows from the basic trigonometric identity $\cos(x) = \sin(x + \frac{\pi}{2})$ and Equation (13).

For Item 2, note that because $\Gamma_n^{\cos}(\,\cdot\,; S, \omega)$ is a sum of shifted versions of $\Gamma^{\sin}(\,\cdot\,; S, \omega)$ such that the interiors of the shifted versions' supports are all disjoint, it is sufficient to upper bound the values of $\Gamma^{\sin}(\,\cdot\,; S, \omega)$. Indeed, inspection of the form of $R_4$ shows that $|R_4(t; S, \omega)| \leq \frac{\pi}{\omega}\min(S, 1-S) \leq \frac{\pi}{2\omega}$. Since $\frac{\pi\omega}{2}|\sin(\pi S)| \leq \frac{\pi\omega}{2}$, the bound follows.

Finally, Item 3 follows because $\Gamma_n^{\cos}(\,\cdot\,; S, \omega)$ is implemented via summation of $4(n+1)$ shifted versions of the function $R_4(\,\cdot\,; S, \omega)$. Since $R_4(\,\cdot\,; S, \omega)$ by definition can be implemented via 4 ReLU functions, we conclude the result. $\qquad\square$

*Proof of Lemma 4.* It is sufficient to show that for $t \in [-r - \frac{\pi}{\beta\omega}, r + \frac{\pi}{\beta\omega}]$

$$\mathbb{E}\Gamma_n^{\cos}(T_k(t; \alpha, \beta); S, \omega) = \cos(\beta\omega t).$$

Fix $t \in [-r - \frac{\pi}{\beta\omega}, r + \frac{\pi}{\beta\omega}]$. By definition, $T_k(\,\cdot\,; \alpha, \beta)$ is supported in $[-2k\alpha, 2k\alpha]$. By our choice of $k$, we have $[-r - \frac{\pi}{\beta\omega}, r + \frac{\pi}{\beta\omega}] \in\subseteq [-2k\alpha, 2k\alpha]$. Let $t \in [2m\alpha, 2(m+1)\alpha]$ for some $m \in \mathbb{Z}$ such that $-k \leq m \leq k - 1$. We invoke Item 1 of Lemma 1 to show that $T_k(t; \alpha, \beta) = T(t - 2m\alpha; \alpha, \beta)$. Now, $T(t - 2m\alpha, \alpha, \beta) \in [0, \alpha\beta] = [0, \frac{(2n+1)\pi}{\omega}]$. Therefore by Item 1 of Lemma 3,

$$\mathbb{E}\Gamma_n^{\cos}(T_k(t; \alpha, \beta); S, \omega) = \cos(\omega T(t - 2m\alpha; \alpha, \beta)). \tag{14}$$

It is now sufficient to show that $\cos(\omega T(t - 2m\alpha; \alpha, \beta)) = \cos(\beta\omega t)$. We consider two cases:

1) If $t - 2m\alpha \in [0, \alpha]$, then $T(t - 2m\alpha; \alpha, \beta) = \beta t - 2m\alpha\beta$. The LHS of Equation (14) becomes

$$\cos(\omega\beta t - 2m\alpha\beta\omega) = \cos(\omega\beta t - 2m(2n+1)\pi) = \cos(\omega\beta t).$$

2) If $t - 2m\alpha \in (\alpha, 2\alpha]$, then $T(t - 2m\alpha; \alpha, \beta) = (2m+2)\alpha\beta - \beta t$ and hence the LHS of Equation (14) becomes

$$\cos(-\omega\beta t + (2m+2)\alpha\beta\omega) = \cos(-\omega\beta t + (2m+2)(2n+1)\pi) = \cos(\omega\beta t).$$

This completes the proof. $\qquad\square$

## B  Uniform Continuity

We will give a sketch of the proof that any $f \in \mathcal{G}_K$ is uniformly continuous. By definition, there exists a finite complex measure $\mu$ over $\mathbb{R}^d$ such that $f(x) = \int \exp(i\langle\xi, x\rangle)\mu(d\xi)$ for every $x \in \mathbb{R}^d$. Applying Hahn-Jordan decomposition theorem and Radon-Nikodym theorem, we conclude that $\mu(d\xi) = \exp(i\theta(\xi))|\mu|(d\xi)$ for some finite measure $|\mu|$ called the total variation measure of $\mu$. Therefore, for arbitrary $x, y \in \mathbb{R}^d$ with $x - y = \delta$.

$$\begin{aligned}
|f(x) - f(y)| &= \left| \int (\exp(i\langle\xi, x\rangle) - \exp(i\langle\xi, y\rangle))\mu(d\xi) \right| \\
&= \left| \int (\exp(i\langle\xi, x\rangle) - \exp(i\langle\xi, y\rangle))\exp(i\theta(\xi))|\mu|(d\xi) \right| \\
&\leq \int \left| \exp(i\langle\xi, x\rangle) - \exp(i\langle\xi, y\rangle) \right| |\mu|(d\xi) \\
&= \int \left| \exp(i\langle\xi, \delta\rangle) - 1 \right| |\mu|(d\xi) \\
&\leq \int 2\min(1, |\langle\xi, \delta\rangle|)|\mu|(d\xi) \\
&\leq \int 2\min(1, \|\xi\|\|\delta\|)|\mu|(d\xi) \\
&:= I(\|\delta\|) \tag{15}
\end{aligned}$$

By dominated convergence theorem, we conclude that $\lim_{\|\delta\| \to 0} I(\|\delta\|) = 0$. Since $I(\|\delta\|)$ depends only on $\|x - y\|$ and not on $x, y$, we conclude that $f$ is uniformly continuous.