[Reviews · NeurIPS 2020]

Review 1

Summary and Contributions: The paper studies the expressiveness (representational capacity) of feed-forward fully connected deep neural networks with ReLU activation. It defines a function class G_K which includes those functions whose Fourier transform decays quickly enough. For larger K the requirement from the decay is more relaxed, and so the series G_1, G_2, G_3, ... is increasing. Given a value for K and a function f in G_K, an upper bound on the approximation error (in L^2 sense) achievable by a network with D layers and N_0 hidden units is established. Loosely speaking, this upper bound scales as (1/N_0)^{D/K}. It is then shown that for the case of univariate functions (input dimension 1), there exist a function in G_K for which the upper bound is tight (up to constants). From a technical perspective, the central idea is to view a function via its inverse Fourier transform, and then approximate cosine components, followed by frequency multiplication via triangle waveforms (which has been done in previous work).

Strengths: Expressiveness of neural networks is a central topic in the theory of deep learning, and the models analyzed in this work (feed-forward neural networks with ReLU activation) are highly relevant to practice. The paper is well written, mathematically solid (I did not verify all details but this is my impression), and despite its technical nature delivers core ideas behind the proofs in a simple and accessible manner.

Weaknesses: My main critique on this work relates to its significance. It is unclear to me why the function class G_K is interesting, and what being able to represent it means. One could view this function class as a means for establishing depth separation (advantage of depth to expressivity), but the proven theorems can be used to establish such result only for univariate functions (input dimension 1). Moreover, as far as I could tell, the lower bound one would obtain for the size a depth D' network required in order for it to be able to match a depth D network with N_0 hidden units, scales as N_0^{D/D'}, which to my knowledge does not go beyond existing literature. My personal opinion is that the practical relevance of this work is questionable. However, since this opinion is largely subjective, and the paper is of high technical quality, I prefer to let NeurIPS readership be the judge of that. === UPDATE FOLLOWING AUTHOR FEEDBACK === I have thoroughly read the authors' feedback, as well as the other reviews. My review of the paper still stands --- I think it is a strong technical contribution with limited significance to practical machine learning. I agree with R3 that further discussion about the significance of the function classes G_K would be very helpful, and with R4 that experiments can also contribute. It is true that those will involve factors beyond just representation (namely, optimization and generalization), but if the paper carries practical relevance some form of empirical demonstration should be possible. With regards to my comment on results being known in the literature, my intent was not that the specific lower bound established in the paper is known, but rather that the depth separation result it induces is not materially stronger than known depth separation results, e.g. those of [23].

Correctness: Yes (though I did not verify all technical details)

Clarity: Yes

Relation to Prior Work: Relation to prior work is good enough in my opinion, but could be improved. I would be interested in a specific account for the technical differences between this work and existing literature. For example, it relies on results from [23] but does not specify where exactly the line passes between that work and the current. I also recall that [22] uses techniques relating to Fourier transform, and would be interested to understand the exact differences. Overall, as this work is largely focused on technical matters, I think an additional section (possibly in an appendix) specifying the exact distinction in terms of mathematical techniques would be helpful.

Reproducibility: Yes

Additional Feedback: I recommend to the others not to shy away from depth separation, and derive a formal result of that type, including comparison to prior work. I believe many readers (myself included) would view that as a more meaningful result than being able to represent G_K. Minor comments: * Line 41: I think the word "on" should be removed. * In Section 2, the use of both n_i and d_i to refer to hidden layer widths is confusing. I suggest choosing just one set of symbols. * In line 99, I think it should be R^d to R, not R^d to R^d. * In line 133, is the constant 2 needed? Seems like a typo to me. * Line 136: "bounds" should be "bound". * In line 158, I recommend writing that l is a natural number.


Review 2

Summary and Contributions: Depth separation is an important open problem in deep learning theory that aims to give a nice description of the role of depth such as to characterize the function class G where depth D neural networks can effectively span the space but depth D’ networks where D’<D cannot. Several previous studies have shown a fast approximation rate of deep networks for very smooth functions, but recently Bresler and Nagaraj revealed that even a single layer network can also achieve a similar rate for such very smooth functions, and thus the depth separation problem is still unsolved. The main contributions of this study are to present a filtration fo function classes G_K and provide both upper and lower bounds of approximation rates for depth D and hidden units number N. (To be precise, the lower bound is shown only for the case when input dimension d=1.) The main technique is a combination of Barron’s scheme to upper bound the approximation rate of the so-called Barron class functions for single-layer networks, Yarotsky’s scheme to convert a multiple-ReLU-layers into an equivalent single-ReLU-layer, and Telgarsky’s scheme to lower bound the approximation rate.

Strengths: The Barron class and Barron’s scheme to derive a dimension independent approximation rate are well-investigated in the context of approximation theory. One of the novelties of this study is to focus on the less-smooth classes, i.e. W^{1,\alpha} with \alpha (= 1/K) \in (0,1), and introduce a new filtration G_K where the order K explicitly bridges the less-smoothness \alpha and the depth D. Moreover, the approximation rate naturally expresses our intuitive understanding of the role of depth: the deeper, the better.

Weaknesses: The authors provide the lower bound for the case when d=1. Can we further provide a lower bound for the cases d >1? I think a lower bound typically depends on the dimension, though.

Correctness: Correct

Clarity: Very clear. Compared to the previous works by Barron (1993), Yarotsky (2018) and Telgarsky (2016), which are dense; this manuscript is simple and clear. I think this manuscript will be a good introduction for the beginners.

Relation to Prior Work: Well reviewed.

Reproducibility: Yes

Additional Feedback: === UPDATE AFTER AUTHOR RESPONSE === I have read the author’s responses and other reviewers’ comments, and I would keep my score as is. I think this could be published at neurips. I agree that several techniques are borrowed from famous works, but I believe that providing an explicit formulation like G_K is new. Hence, the bounds are not just a rate with undescribed/insanely large coefficients. I do not think that the space G_K to be pathological, because this is almost (not exactly, though) equivalent to the L^1-Sobolev space with order 1/K, i.e. W^{1,1/K}. Recently, “the solutions obtained by SGD” draw much attention, but I think this is simply an independent question of expressive power. By the way, I have some questions in the response letter. 1. “This work focuses on dimension-free bounds and therefore the lower bounds are also given in terms of the ‘Fourier norms’ which are dimension-free.” --- I am not sure if this is really correct for dimension d>1. Lower bounds are typically suffered from dimension. I think this is optional (and need not to be proved in the final version) but stating it without proof sounds a little loose. 2. “explain features of GK like 1) uniform continuity” --- I am not sure if this is true because the authors assume F(\xi)d\xi to be a finite complex measure. (a) If F is an L^1-function, then it is true. (b) According to Bochner’s theorem, the Fourier transform of finite Borel measure is continuous (and positive definite), but I do not have any textbook that claims it is ‘uniformly continuous’.


Review 3

Summary and Contributions: This paper considers the representation power of ReLU neural networks, proving a tight dimension-free upper bound on the depth needed to approximate functions with finite ‘Fourier norm’. A corresponding lower bound is given by constructing a hard case, showing that the dependence on depth is sharp.

Strengths: I think the idea of this paper is quite smart: a non-periodic function on a compact set can be extended to the whole domain as a periodic one, then we can use its Fourier series to rewrite it as the sum of cosines! Neural networks are known to be able to represent ‘dog-teeth’ functions efficiently, so the approximation of cosines is straightforward. The bound is tight and intuitively show how depth affects the neural network’s approximation power. It’s also free of input dimension.

Weaknesses: As the authors admitted, the use of function classes G_K with bounded Fourier norm needs more justification. What’s its relationship to other common function classes? And how to judge which G_K a given function belongs to? It would be better if the authors can provide two or three specific examples if it’s hard to characterize generally.

Correctness: I did go through the proofs but not every detail. The results should be correct, and they are actually not surprising given the clear idea behind them.

Clarity: This paper is well-written and very easy to follow.

Relation to Prior Work: The discussion part is comprehensive.

Reproducibility: Yes

Additional Feedback: === UPDATE AFTER AUTHOR RESPONSE === I would keep my score after reading authors' feedback along with other reviews.


Review 4

Summary and Contributions: This paper derives a representation theorem for a class of functions with varying smoothness which is based on a careful approximation of sinusoidal functions using ReLU nonlinearities. The theorem shows that deep networks have qualitatively more representation power for this class of signals.

Strengths: The paper is written very clearly. The authors explain their arguments in a very accessible way. The result is interesting, although purely from a theoretical point of view.

Weaknesses: The scope of the contribution is somewhat limited (although interesting nonetheless). The authors do not discuss the limitations of the result when explaining the behavior of neural networks in practice (mainly that the network parameters are learned, so it is unclear that the representation is attainable via sgd). This would have made the paper potentially more interesting to a more general audience. In the same spirit, the authors do not provide any computational experiments showing that the effect they capture is observable when calibrating the networks using sgd. After reading the author's response, I am keeping my score.

Correctness: Yes, although I could not verify the calculations in detail due to time constraints.

Clarity: Yes, very well written.

Relation to Prior Work: Yes.

Reproducibility: Yes

Additional Feedback:

[Author Response · NeurIPS 2020]

We thank the reviewers for their valuable and constructive feedback. We will incorporate the suggestions in the next
revision. Our responses to some specific comments are below.

**Common Comments:**

• **Limitation of Lower Bounds**: We argue that our lower bounds are strong since they show that even in the case of
a 1-D function (which is the simplest case for representation), the lower bounds match the upper bounds. A one
dimensional function can be embedded in $\mathbb{R}^d$ by considering $g(x) := f(\langle e_1, x \rangle)$ and the lower bounds follow. A
similar trick is used in Telgarsky's work which shows depth separation.
This work focuses on dimension-free bounds and therefore the lower bounds are also given in terms of the 'Fourier
norms' which are dimension-free. The Fourier norms may themselves depend on the dimension for specific function
classes like in Example 1 of Section 2, but the dimension dependence (if at all) is through the Fourier norms.

• **Regarding the Function Spaces**: It is indeed the case that the function spaces $\mathcal{G}_K$ are not easily characterized in
terms of standard function spaces. Some intuition has been given in Examples 1 and 2 of Section 2. Similar spaces
are used in classic works of Barron et. al to show the first quantitative representation theorems for neural networks
and such function classes are well investigated in the approximation theory literature. We will add more references
to these works and further explain features of $\mathcal{G}_K$ like 1) uniform continuity and 2) density of a subspace of $\mathcal{G}_K$ in
$L^p(\mathbb{R}^d)$. The optimality of our results suggests that the spaces $\mathcal{G}_K$ are natural in the study of neural networks.

**Reviewer Specific Comments:**

**Reviewer 1**: See items 1 and 2 in common comments. The reviewer claimed that the lower bounds are already known in
the literature. We could not find the references to such results. We request the reviewer to share any relevant references
regarding this.

• **Regarding Literature Review** [23] considers depth separation - i.e, they construct functions which are outputs of a
$D$ layer network with a small number of neurons but need a huge number of neurons to represent with fewer layers
($D' \ll D$). We use the basic idea of bounding oscillations to show a quantitative lower bound on accuracy when
representing certain functions in the class $\mathcal{G}_K$.
[22] uses a careful manipulation of the Parseval-Plancherel formula, radial symmetry and the (lack of) support of
Fourier transforms of neural network outputs to obtain a specific distribution and a specific function under which 2
layer networks cannot approximate radial functions easily. This is different from the present work which obtains a
sampling procedure via the Fourier distribution in order to show a strong approximation result.
We will make these distinctions clear in the next version of our manuscript.

**Reviewer 2**:See item 1 in Common Comments.

**Reviewer 3**: See item 2 in Common Comments.

**Reviewer 4**:

• **Regarding Relevance of the Work**:
We did not discuss the training aspects of neural networks because we only concentrated on the representation power
of deep networks. Based on current literature, theoretically understanding the SGD based training of deep neural
networks appears to be a very hard problem. Nevertheless, representation results are important and interesting, and
the long line of work on representation power of neural networks shows continued interest in the topic. Precise and
optimal bounds on the expressive power of neural networks are algorithm independent and fundamental properties
of neural networks. We believe it is essential to understand these aspects in order to understand specific training
algorithms like SGD. For instance, in the paper https://arxiv.org/pdf/2001.04413.pdf, a precise upper bound on
representation power of kernels is used to show that 3 layer networks outperform 2 layer networks in certain learning
tasks via SGD type algorithms.

• **Regarding computational experiments**: Computational experiments of this nature would unfortunately be mislead-
ing because they would compare training via SGD to a purely representation result. This, in our opinion, cannot be
justified easily and does not contribute to the results established in the paper.

[Meta-Review · NeurIPS 2020]

Four knowledgeable referees rate this submission 7,7,6,6. The article presents results on the representational power of neural networks and depth separation. The referees agree that this is a topic of interest to the community and that this is a strong technical contribution albeit with limited significance to practical machine learning. They lean towards accept. I agree with the referees and hence I am recommending accept, but would like to encourage the authors to address the comments and suggestions of the referees in the preparation of the final manuscript, in particular from R1, R3, R4 about the significance.